# Comparison of Automated and Traditional Western Blotting Methods

**DOI:** 10.3390/mps6020043

**Published:** 2023-04-20

**Authors:** Aino Sormunen, Emma Koivulehto, Kari Alitalo, Kalle Saksela, Nihay Laham-Karam, Seppo Ylä-Herttuala

**Affiliations:** 1A. I. Virtanen Institute for Molecular Sciences, University of Eastern Finland, FI-70211 Kuopio, Finland; 2Translational Cancer Biology, University of Helsinki, FI-00014 Helsinki, Finland; 3Department of Virology, University of Helsinki, FI-00014 Helsinki, Finland; 4Heart Center and Gene Therapy Unit, Kuopio University Hospital, FI-70029 Kuopio, Finland

**Keywords:** Western blotting, automation, protein analysis, JESS Simple Western™, iBind™ Flex, SARS-CoV-2

## Abstract

Traditional Western blotting is one of the most used analytical techniques in biological research. However, it can be time-consuming and suffer from a lack of reproducibility. Consequently, devices with different degrees of automation have been developed. These include semi-automated techniques and fully automated devices that replicate all stages downstream of the sample preparation, including sample size separation, immunoblotting, imaging, and analysis. We directly compared traditional Western blotting with two different automated systems, iBind™ Flex, which is a semi-automated system designed to perform the immunoblotting, and JESS Simple Western™, a fully automated and capillary-based system performing all steps downstream of sample preparation and loading, including imaging and image analysis. We found that a fully automated system can save time and importantly offer valuable sensitivity. This is particularly beneficial for limited sample amounts. The downside of automation is the cost of devices and reagents. Nevertheless, automation can be a good option to increase output and facilitate sensitive protein analyses.

## 1. Introduction

Western blotting (WB) was invented over four decades ago [1,2] and is still one of the most utilized analytical techniques in multiple different fields of research and diagnostics. The key benefit of WB is the ability to identify specific target proteins from complex protein samples, such as tissue or cell lysates. This is normally performed by separating proteins using gel electrophoresis, transferring them to a membrane, and probing the target proteins with specific antibodies (Abs), followed by imaging of the Ab binding either via an enzymatic color reaction, a fluorescent label, or an enzyme-activated chemiluminescence signal [3].

Despite the popularity of the traditional WB as a key method, it also has its problems. Some of the difficulties related to WB are the problematic detection of low-abundance proteins, the need for large amounts of proteins, dependence on the availability of specific Abs, and potential unreliability in the normalization of the target proteins. In addition, difficulties in reproducibility can sometimes be faced in WB either due to the variations in the multiple different steps of the procedure or the use of different reagents between laboratories. As there is no real agreement on what comprises reproducibility in WB, this can cause problems [4,5,6]. Likewise, the time needed for the entire procedure, which can vary between 1 and 3 days, can also be problematic in time-limited projects.

To overcome these pitfalls, improvements in the various aspects of WB have been implemented. These have extended over the full range of the procedure from the availability of precast acrylamide gels to instruments that complete the entire process. In particular, companies have developed different automated devices for WB to increase reproducibility and sensitivity and decrease performance time. Two examples of such devices are iBind™ Flex, developed by Thermo Fisher Scientific, Waltham, MA, USA, and JESS Simple Western™, developed by ProteinSimple^®^, Bio-Techne, Minneapolis, MN, USA.

iBind™ Flex is an automated WB device designed to perform all steps of the immunoblotting procedure, including the blocking, primary and secondary Ab incubations, as well as the in-between washes [7]. This then leaves the sample preparation, gel electrophoresis, membrane transfer, and imaging to the user. The instrument utilizes sequential lateral flow to process the blocking, Abs, and washing solutions. This technique allows all the solutions to flow to the membrane in a timely and controlled fashion, reducing the hands-on time used for the whole procedure (3 h). In addition, although the Ab concentration required is often higher for this instrument, the volumes used are considerably smaller, thereby reducing the required amounts of Abs [7].

JESS Simple Western™, on the other hand, is automated to perform all the steps of protein size separation and immunoblotting that are associated with WB [8]. In this instrument, capillaries replace the acrylamide gel and the transfer membrane, such that samples are loaded, size separated, and immunoblotted in the capillaries. The proteins in the samples separate in the capillaries by migrating through stacking and separation matrices and are immobilized to the capillary walls via photoactivated capture chemistry. This then allows for the immunoblotting to take place in the capillaries to detect specific immobilized proteins. The immunoblotting follows a standard regime of primary and (labeled) secondary Abs followed by a detection system and imaging. Indeed, one of the advantages of JESS Simple Western™ is that the imaging of either a fluorescent tag or a chemiluminescence signal, as well as an analysis of the images, are also completed within the run. This way, only the sample preparation, along with pipetting to a special plate, are left to the user. This method, thus, reduces the time and amount of sample used for the entire procedure, and greater reproducibility can be achieved through the automation of all critical steps [8,9].

In this report, we provide a direct comparison of the automated WB techniques, iBind™ Flex and JESS Simple Western™, with traditional WB.

## 2. Materials and Methods

**Cell lysates.** Samples based on transfected 293T cells were generated for this study. The cells plated in 6-well plates at 1.2 × 10^6^ cells/well were transfected, utilizing Lipofectamine™ 3000 (Invitrogen, Waltham, MA, USA), with adenoviral vector plasmids (2.5 µg), expressing the receptor binding domain (RBD) of severe acute respiratory syndrome coronavirus 2 (SARS-CoV-2) in different-sized constructs (RBD1–4). The range of protein products varied from ~36 kDa to 215 kDa. Control lysates were generated from 293T cells transfected with adenoviral vector plasmid overexpressing vascular endothelial growth factor (VEGF; 2.5 µg) construct. Two days after transfection, 293T cells were lysed with radioimmunoprecipitation assay (RIPA) buffer [10] at 4 °C for 30 min, after which the lysates were cleared of cell debris by using centrifugation at 2000× *g* for 5 min. The protein concentrations of the samples were quantitated using Pierce™ BCA Protein Assay Kit (Thermo Scientific, Waltham, MA, USA).

**Traditional WB.** Standard sodium dodecyl sulfate–polyacrylamide gel electrophoresis (SDS-PAGE) protocol, utilizing 4–20% gradient pre-cast gels (Mini-PROTEAN^®^ TGX Stain-Free™ Gels; Bio-Rad, Hercules, CA, USA), was applied for separation of the proteins. A total of 10 µg, 2.5 µg, and 1 µg of total protein from each sample was loaded onto gels in 1x Laemmli sample buffer [10]. Following electrophoresis, the separated proteins were transferred to 0.2 µm nitrocellulose membranes (Bio-Rad) using Trans-Blot Turbo Transfer System (Bio-Rad).

Nonspecific Ab binding was blocked using 5% bovine serum albumin (BSA; Sigma, St. Louis, MO, USA) in Tris-buffered solution with 0.05% Tween (TBST; TBS-Tween tablets; Medicago, Quebec City, QC, Canada). Polyclonal primary Ab, rabbit IgG anti-SARS-CoV-2 RBD (1:2500, Sino Biological: 40592-T62, Beijing, China, RRID: AB_2927483), was diluted in 5% BSA in TBST (10 mL) and incubated overnight at 4 °C on a shaker (90 rpm). Prior to adding the secondary Ab, the membranes were washed four times using TBST solution for 10 min, but discarding the first wash immediately, to remove unbound primary Abs. Secondary Ab, horseradish peroxidase (HRP)-conjugated goat anti-rabbit IgG (1:2000, Invitrogen: 31460, RRID: AB_228341), was diluted in TBST (10 mL) and allowed to bind at room temperature for 2 h on a rocker. Membranes were washed as above. Pierce ECL Plus Western Blotting substrate (Thermo Fisher Scientific) was diluted according to the manufacturer’s instructions and added to the membrane for 3 min. Excess substrate was removed prior to imaging on Bio-Rad ChemiDoc Imaging System.

To verify similar protein loading and to normalize protein amounts in the lysates, the membranes were stripped and reblotted with Ab to detect a housekeeping protein, Glyceraldehyde 3-phosphate dehydrogenase (GAPDH). Membranes were incubated in stripping buffer (2% SDS, 0.06 M Tris-HCl, pH 6.8, 0.7% β-mercaptoethanol, H2O) at 65 °C for 30 min with shaking. Membranes were rinsed under running mH2O for 2 min, after which they were washed with TBST as before. After stripping, the membranes were blocked, immunoblotted with rabbit mAb anti-GAPDH diluted 1:1000 (Cell Signaling Technologies: 2118, Danvers, MA, USA, RRID: AB_561053), and imaged on Bio-Rad ChemiDoc Imaging System as detailed above.

**iBind™ Flex.** For testing the iBind™ Flex Western System (Fisher Thermo Scientific), the SDS-PAGE protocol, gel transfer, stripping for reblotting, and blocking were applied as detailed above for the traditional WB. The iBind™ Flex device and iBind™ Flex cards were used for immunodetection according to the manufacturer’s protocol. Primary Abs (2 mL), rabbit IgG anti-SARS-CoV-2 RBD (1:2500, Sino Biological: 40592-T62, RRID: AB_2927483), rabbit mAb anti-GAPDH (1:1000, Cell Signaling Technologies: 2118, RRID: AB_561053) and secondary Ab (2 mL), and HRP-conjugated goat anti-rabbit IgG (1:400, Invitrogen: 31460, RRID: AB_228341) were used. RBD and GAPDH were detected sequentially following stripping of the membranes after the first protein detection. Membranes were imaged using Bio-Rad ChemiDoc Imaging System as detailed above.

**JESS Simple Western™.** To analyze lysates on the JESS Simple Western™ instrument (ProteinSimple^®^, Bio-Techne, Minneapolis, MN, USA), samples were diluted to total protein concentrations of 100 ng/µL, 50 ng/µL, 25 ng/µL, and 12.5 ng/µL in 1x fluorescent master mix (EZ standard pack I; ProteinSimple^®^), and 3 µL was added per well. The polyclonal rabbit IgG anti-SARS-CoV-2 RBD (Sino Biological: 40592-T62, RRID: AB_2927483) was used at 1:50 and the monoclonal rabbit anti-GAPDH (Cell Signaling Technologies: 2118, RRID: AB_561053) at 1:10, diluting in Milk-Free Ab diluent (Bio-Techne) and adding 5 µL per well. Overall, the manufacturer’s protocol was followed with modifications only to the volumes applied. The secondary Ab (anti-rabbit HRP) and enhanced chemiluminescence (ECL) reagents were used according to the kit’s instructions (anti-rabbit detection module chemiluminescence; ProteinSimple^®^, Bio-Techne). The Ab diluent, washing buffer, plates and capillary cartridges used were derived from the 12–230 kDa separation module (ProteinSimple^®^, Bio-Techne). To allow sequential detection of both rabbit primary Abs, a RePlex™ reagent kit was used (ProteinSimple^®^, Bio-Techne).

**Image analysis.** Chemi high-sensitivity images taken with a ChemiDoc imaging system following 7.8 s (RBD) and up to 62 s (GAPDH iBind™ Flex) exposure with 4 × 4 binning were analyzed with Image Lab 5.1 software (Bio-Rad). From these images, automatic bands were detected with setting of high sensitivity and lane background was subtracted with disk size of 10. The band volume (Int) from non-saturated images were used to compare expression of different proteins.

JESS Simple Western™ data were analyzed using Compass for Simple Western software. Images from the high dynamic range 4.0 were used for the analysis, and peaks were automatically detected. Both peak height and area were analyzed.

## 3. Results

In order to compare different WB methods for protein analysis, we generated cellular lysates from transfected 293T cells expressing transgenes of different sizes, all containing the SARS-CoV-2 RBD to enable detection with the same primary Ab. The different-sized proteins generated were ~36 kDa, 66 kDa, and 215 kDa, which were all detected by traditional WB (Figure 1a), along with the housekeeping protein, GAPDH (42 kDa). Total protein amounts from 10 µg to 1 µg were used for each of the RBD lysates to assess the sensitivity of the method. Both at 10 µg and 2.5 µg of total protein, all RBD proteins were detected, with, naturally, the 10 µg bands being brighter. The RBD2 and RBD3 bands seemed fainter compared to RBD1 and RBD4, possibly due to their larger size, with some of the protein not running optimally (observed smearing). At 1 µg of total protein, the bands from overexpressed RBD proteins could still be detected. Similarly, GAPDH was efficiently detected at 10 µg, with the bands being similar in all samples, indicating equal protein loading. At both 2.5 µg and 1 µg, the expression was lower, but still detectable. A comparison of the quantitation of RBD and GAPDH band volumes from independent experiments (see Appendix A for blot images) is represented in Figure 1b.

Membranes of the same lysates processed using the iBind™ Flex also demonstrated the RBD (1–4) proteins and GAPDH (Figure 2a). While the iBind™ Flex showed bands of good quality in the lanes loaded with 10 µg of total protein, the sensitivity was lower, as often either RBD or GAPDH proteins were undetected at 1 µg of total protein (Figure 2a and Appendix A) using the same exposure conditions or even higher exposures of up to 62 s. Matching traditional WB only required exposure times of 4.4 to 7.8 s. The difference in sensitivity between traditional WB and iBind™ Flex immunoblotting was consistently observed over repeated experiments (Appendix A). The band volumes for RBD and GAPDH bands from three independent experiments are represented in Figure 2b.

The band volumes of the RBD1–4 bands were normalized to those of GAPDH, from both the traditional WB and iBind™ Flex membranes (Figure 3). In traditional WB, the pattern of expression was replicated at different lysate amounts used, and this was so for all three protein amounts in the represented experiment (Figure 3a) or in at least two out of three protein amounts in the mean data from independent experiments (Figure 3b). In contrast, the normalized data following iBind™ Flex processing did not demonstrate consistent expression profiles for the different RBD proteins at the different protein concentrations either in individual (Figure 3a) or combined experiments (Figure 3b). This analysis from the iBind™ blots was hampered by the lack of detection of either RBD or GAPDH at lower concentrations.

In addition to the traditional WB method and iBind™ Flex, the lysates from the transfected 293T cells were processed using the automated device JESS Simple Western™. Since both primary Abs were rabbit, a RePlex™ assay was utilized, which is analogous to stripping and reblotting in the traditional WB. While the RBD-containing proteins were clearly visible at all concentrations, the housekeeping protein GAPDH was very faint at 25 ng/µL and not always detectable at 12.5 ng/µL (Figure 4a and Appendix A), highlighting the difference between overexpression of a transgene and the expression of an endogenous gene. It was also possible to visualize some smaller-sized products of the transgenes, which may be either different glycosylation forms of the proteins or partial products. Interestingly, only one of these was visible in the traditional WB and none were visible in the iBind™ Flex.

The binding of the Abs in the capillaries is visualized in different formats in the Compass for Simple Western analysis software, either as a traditional blot-like image in “lane view” (Figure 4a) or as an electropherogram, from which peak heights and areas can be determined. An example of this is given in Figure 4b, which clearly demonstrates the peaks of the RBD proteins at 42 kDa, 66 kDa, and ~215 kDa, as well as the peaks of GAPDH at 42 kDa. The molecular weights of the peaks were calculated from a comparison to the ladder. In Figure 4c,d and e the peak areas for RBD and GAPDH proteins and the normalized RBD (RBD/GAPDH) from three independent experiments are represented, respectively. For this analysis, we included peak areas from all samples, including those that exceeded the recommended peak height cutoff of 600,000 (higher concentration samples), to observe the potential variation across all concentrations. Similar to traditional WB, using JESS Simple Western™, the patterns of peak areas for the RBD1–4 samples were consistent across all the concentrations used, including the lowest concentration of 12.5 ng/µL. The detection of protein at 12.5 ng/µL equates to the detection of total protein of 0.375–0.625 ng if taking into consideration the sample volume taken up into the columns is 30–50 nL; this highlights the sensitivity of the automated WB over the other methods tested here. However, unlike in both the traditional WB and iBind™ Flex, higher-molecular-weight proteins (RBD2–3) were detected at higher levels than RBD1 and 4. This could possibly be due to a large amount of protein and/or the suboptimal running of the larger-sized RBD2 and RBD3 in the traditional WB, which resulted in more diffuse electrophoresis. Alternatively, in JESS Simple Western™, these higher-molecular-weight proteins did not separate adequately and so were observed as a single peak. Cross-reactivity with the internal fluorescent markers, which are used to monitor electrophoresis through the capillary, can also occur. However, the absence of peaks at the high molecular wight in RBD1 and 4 as well as control wells showed that this is not the case (Figure 4a).

## 4. Discussion

In this report, we compared traditional WB with a semi-automated system for immunoblotting, iBind^TM^ Flex (Thermo Fisher Scientific), and a fully automated system, JESS Simple Western^TM^ (ProteinSimple^®^, Bio-Techne). While each of these systems might have its benefits, they also have some disadvantages, and all aspects must be considered, including costs, time, and sensitivity, among others. Based on these results, traditional WB seems to be more sensitive than iBind™ Flex. Traditional WB was able to mostly replicate the results after normalization at a large range of total protein amounts (10–1 µg), unlike iBind™ Flex. Furthermore, the smaller products, which are likely to be different glycosylation forms, were visible in traditional WB and JESS Simple Western™ but not in iBind™ Flex. However, it appeared that iBind™ Flex can also perform well but was more Ab-dependent, as the anti-GAPDH Ab detected proteins also at 1 µg, unlike the anti-RBD Ab. Thus, for reproducibility and, potentially, for sensitivity, traditional WB is probably a better option. If the best sensitivity is the goal, JESS Simple Western™ is the best choice because proteins were detected at a lysate concentration of 12.5 ng/µL. Because only approximately 30–50 nL of the 3 µL sample volume is actually processed through the columns, then the detection is at a total protein amount of <1 ng. Furthermore, JESS Simple Western™ also consistently detected the same pattern of expression across different samples at all the concentrations tested. Hence, in addition to sensitivity, it also demonstrated reproducibility.

One clear benefit of the automation of WB, whether it is partial automation of only the immunoblotting (iBind™ Flex) or of the complete procedure (JESS Simple Western™), is time-saving. Whereas traditional WB can take 1–3 days from loading samples to imaging, using iBind™ Flex, this can be completed in 1 day, and with JESS, in 3–5 h, depending on the assay type. JESS also includes automated analysis of the detected peaks and is, by far, the most time-efficient procedure. Of course, time-saving should not come at the cost of compromised quality, which can be the case with the iBind™ Flex. However, this is Ab-dependent because we observed efficient detection using iBind™ Flex with other Abs that are not included in this work. Hence, validation of the Abs is necessary for all methods. This is particularly true for the JESS Simple Western™, where testing of both the sample concentration and the Ab dilution is required to ensure a specific and reliable quantitative detection, despite a large linear dynamic range [8].

Compared to traditional WB, the main advantage of the automated JESS Simple Western™ is the ability to assay limited sample amounts [9]. This is due to the high sensitivity and small volume requirements, which enable analysis from limited precious samples, such as human samples or small animal tissues. Ordinarily, for traditional WB, samples of 1–20 µg of total protein are used, while in JESS, 3 µL of 50–12.5 ng/µL sample concentrations were sufficient to detect the proteins. The other advantage of JESS is the ease of multiplexing with the availability of chemiluminescence, near-infrared, and infrared detection. However, chemiluminescence has the highest sensitivity. Thus, sequential detection with RePlexing is advisable to take advantage of the chemiluminescence sensitivity for all detected proteins. Of course, in the case that the target and housekeeping proteins are of different molecular weights, which can be sufficiently resolved, and the cross-reactivities of the Abs allow, simultaneous detection of the target and housekeeping proteins can be achieved by mixing the primary Abs, which would avoid run-to-run variability.

Normalization of the protein of interest and accounting for variations in sample loading are also very important in all WB-based methods. This is often performed by the detection of a housekeeping protein in the same blot. However, depending on treatments, the expression of the housekeeping protein can vary, which can be problematic for normalization. In this case, the quantitation of the total protein in the lane can be more reliable. Of course, traditional silver or Coomassie stains can be employed to stain the acrylamide gel, but this cannot then be used to transfer to the membrane. Instead, Ponceau staining of the membranes or the trihalo compounds, which can be incorporated into the acrylamide gels, can be used. In particular, the trihalo compounds are of interest as these can covalently bind tryptophan and have enhanced fluorescence when exposed to UV light, which can also be reactivated multiple times for visualization in the gel and membrane [11,12]. Similarly, in JESS Simple Western™, total proteins can be quantitated using a propriety fluorescent compound that binds all amine groups and is detected on a separate channel. This can expand the normalization options [13].

Another consideration in the comparison of the traditional WB and the automated systems is the Ab consumption. In this study, the primary Ab consumption was essentially similar to the different methods. Although the automated systems require more concentrated Abs, the volumes needed are smaller. Hence, the costs for Ab remain relatively similar between the methods. However, compared to the traditional WB, the downside of the JESS Simple Western™ includes the cost of buying the instrument, as well as the separation modules and reagents. However, some of these costs can be offset by the savings due to reduced hands-on time in running and analyzing the samples. Additionally, all consumables, such as the capillary cartridges and sample plates are single-use and only available from a single vendor. In contrast, for traditional WB, the gel equipment can be reused, the membrane can be reblotted many times, and reagent availability is much more widespread and, consequently, more price-competitive. Nevertheless, JESS Simple Western™ does facilitate the WB process through good sensitivity and increased throughput and reproducibility.

## 5. Conclusions

WB is one of the most widely applied and valued methods in biological research. Although the traditional method is still commonly used, the shift towards increased automation has also brought improvements. Automation can reduce hands-on and experimental time, as well as increase throughput while maintaining high sensitivity. Overall, WB by the traditional method or iBind™ Flex can be more cost-effective and require less validation, but automated systems, such as JESS Simple Western™, allow easier analysis and quantification of the proteins, as well as higher reproducibility of the results. Likewise, the small sample volumes required are an advantage for limited or precious samples, making their use for protein analysis more feasible than with the traditional WB methods.

## Figures and Tables

**Figure 1 mps-06-00043-f001:**
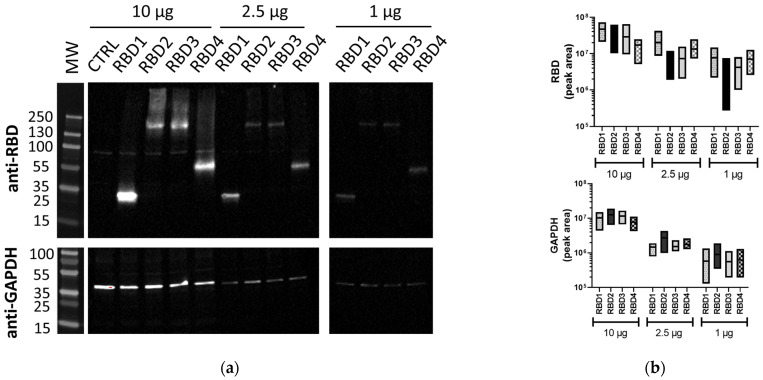
Lysates containing RBD proteins run by traditional WB. (**a**) Image of membranes immunoblotted with anti-SARS-CoV-2 RBD (1:2500) or anti-GAPDH (1:1000) Abs using traditional WB. (**b**) Floating box plots showing the minimum, maximum, and mean of the band volumes for RBD and GAPDH bands for RBD1–4 samples from three independent experiments that were obtained using Image Lab analysis. Lysates (10, 2.5, and 1 µg/lane) were from 293T cells transfected with plasmids expressing different-sized transgenes containing RBD (RBD1–4) or control transgene (CTRL). MW: PageRuler™ Plus Prestained Protein Ladder (Thermo Scientific).

**Figure 2 mps-06-00043-f002:**
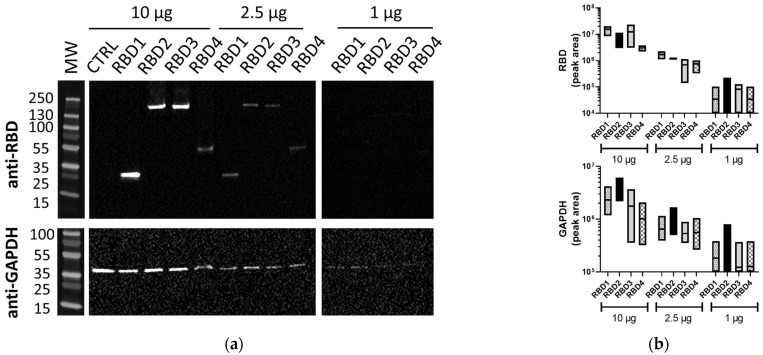
Lysates containing RBD proteins run using iBind™ Flex. (**a**) Image of membranes immunoblotted with anti-SARS-CoV-2 RBD (1:2500) or anti-GAPDH (1:1000) Abs using iBind™ Flex. (**b**) Floating box plots showing the minimum, maximum, and mean of the band volumes from RBD and GAPDH bands for RBD1–4 samples from three independent experiments that were obtained using Image Lab analysis. Lysates (10, 2.5, and 1 µg/lane) were from 293T cells transfected with plasmids expressing different-sized transgenes containing the RBD (RBD1–4) or control transgene (CTRL). MW: PageRuler™ Plus Prestained Protein Ladder (Thermo Scientific).

**Figure 3 mps-06-00043-f003:**
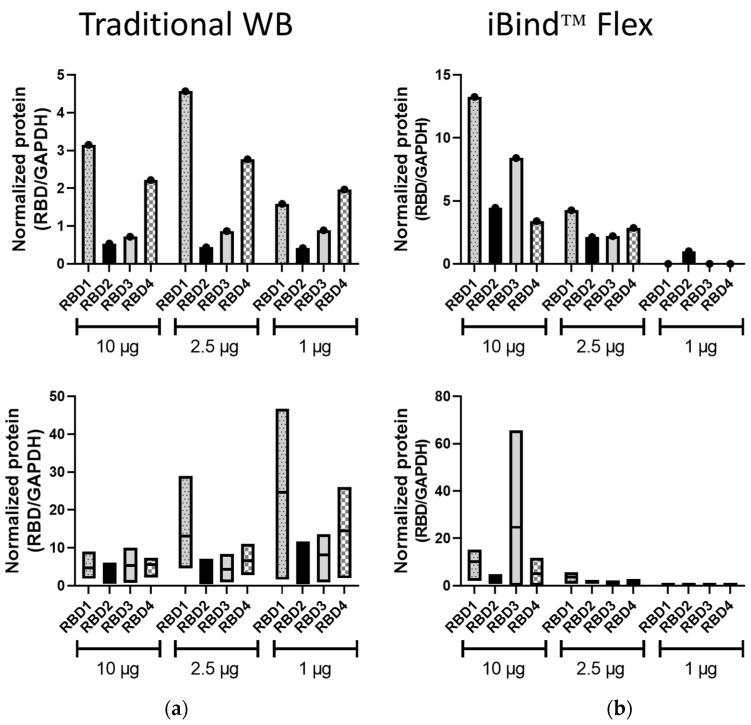
Normalized RBD expression in traditional and iBind™ Flex membranes. RBD (1–4) expression in both traditional WB (**a**) and iBind™ Flex membranes (**b**) was normalized to GAPDH expression using band volume obtained from Image Lab analysis. Top bar diagrams represent quantitation from the representative WBs in Figure 1 and Figure 2. Bottom graphs are floating bar representations (minimum, maximum, and mean) of band volume quantitation from three independent experiments.

**Figure 4 mps-06-00043-f004:**
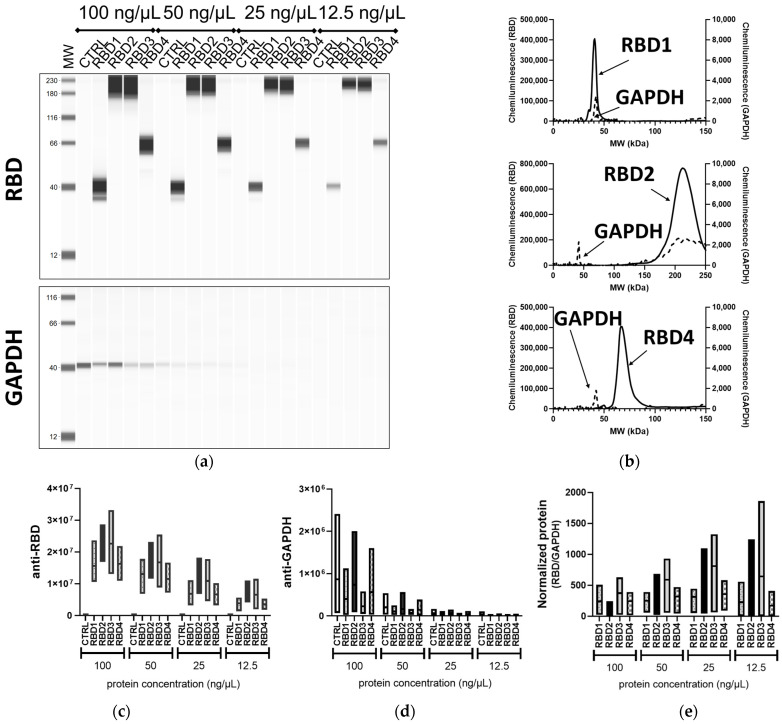
Assessment of transgene expression by JESS Simple Western™. Different concentrations (100, 50, 25, 12.5 ng/µL) of lysates from 293T cells transfected with vectors expressing RBD containing transgenes of different sizes (RBD1–4) or control transgene (CTRL) were processed in the fully automated JESS Simple Western™ device, probed with the SARS-CoV-2 RBD (1:50) and GAPDH (1:10) Abs. (**a**) Portrayal of a traditional blot-like image with a lane view of the samples. (**b**) An electropherogram representation of RBD1, RBD2, and RBD4 (50 ng/µL) samples for both the anti-RBD (solid line) and anti-GAPDH (dashed lines) analysis. Floating bar diagrams (minimum, maximum, and mean) of the peak areas corresponding to RBD (**c**) and GAPDH (**d**) protein from three independent experiments. (**e**) Floating bar diagram (minimum, maximum, and mean; *n* = 3) of the normalized RBD expression calculated from the peak areas of RBD and GAPDH signals.

## Data Availability

No additional data were created or analyzed in this study. Data sharing is not applicable to this article. The authors can be contacted for any further information regarding the data within the article.

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
