# Peer review of "Comparison of Automated and Traditional Western Blotting Methods"

_mps, 2023, doi:10.3390/mps6020043_

Round 1
Reviewer 1 Report
The Authors compared traditional Western blotting with a semi-automated system iBind™ Flex (Thermo Fisher Scientific) and with a fully-automated system JESS Simple Western™ (ProteinSimple®, Bio-Techne). The manuscript is very well-written, the comparisons are clear and very precise. I have not found any factual errors within the body of the manuscript. Regarding the figures, I would rather use boxes with whiskers or scatter dot plots instead of bar charts. The bar charts are commonly used for most scientific reports, but they are statistically inappropriate since they do not show the variability of data points within the sample. From the statistical point of view, the bar charts suggest that the values of the data points within the sample vary from 0 (bottom of the bar) to the value shown by the top of the bar (which is actually the arithmetic mean within the group). I have also found two minor flaws within the article body:
- page 1, line 42: I would rather use the more elegant phrase “multiple different steps” instead of “many different steps”;
- page 7, line 226: the Discussion starts in the following way: “While each of these systems might have their benefits...” – I feel like the introductory sentence is missing here, for example: “In this article we have compared a traditional Western blotting with a semi-automated system iBind™ Flex (Thermo Fisher Scientific) and with a fully-automated system JESS Simple Western™ (ProteinSimple®, Bio-Techne).”
Author Response
First and foremost, we would like to sincerely thank Reviewer 1 for their constructive comments which have improved this manuscript. We were encouraged by the reviewers’ positive assessment of our manuscript and have endeavored to address all the comments.
We thank the reviewer for their helpful suggestion for the data presentation. We have changed the graphs to ‘floating box’ plots that show the minimum, maximum and mean, that demonstrate the variance of the data. We have also added the Western blot analysis images of repeated experiments in the supplementary figures 1 and 2 and have edited the text accordingly on pages 4-7.
We also thank the reviewer for their suggestion for an introductory sentence for the discussion (page 7) and the text on page 1 (line 42), both of which we have changed according to your helpful suggestions.
In addition, during the revision we found a mistake in the labeling of the protein amounts for the traditional WB and iBind™ Flex immunoblotting (10, 1 and 0.1 ug were indicated these should have been 10, 2.5 and 1 ug). We sincerely apology for this mistake which we now have corrected in the text and figure labels (figures 1 and 2)
We hope that we have addressed all the reviewers’ comments satisfactorily and the manuscript is now acceptable for publication.
Reviewer 2 Report
This article by Sormunen et al compares traditional western blotting techniques to newer automated techniques. The paper is generally sound and well written, and I agree with the conclusions that the JESS system might offer advantages due to requiring the smallest amount of sample. however there are a few problems that prevent its publication in current form.
The largest issue is the apparent lack of replication. Critically there is no mention of whether experiments were performed more than once, so at face value this article is asking to be published based on one experiment in each approach. Authors to address this by providing information as to how experiments were replicated and by showing these extra blots in the supplemental information.
I also wonder whether authors had tried increasing the detection time to visualise bands when loading lower protein amounts? This could influence the results as you would likely see improved detection.
Other, less import changes:
Line 75 - how many cells? Was this a dish or 6-well plate etc.
Line 80 - was VEGF-B transfected using the same plasmid amount as the RBD constructs?
Line 88 - what is 1x sample buffer?
Line 93, and throughout - category numbers, and ideally RRID for antibodies should be provided. For example, Invitrogen sells many different goat anti-rabbit HRP antibodies, which one exactly did you use?
Line 110 - needs information about the secondary here too.
Line 115 - needs information about the re-probing as currently sounds like the two primary antibodies were used together which is inappropriate.
Line 234 - a comparison as to the mass of protein loaded in each technique would be useful to help compare between techniques.
Author Response
First and foremost, we would like to sincerely thank the reviewers for their constructive comments which have improved this manuscript. We were encouraged by the reviewers’ positive assessment of our manuscript and have endeavored to address all their comments.
We are grateful to the reviewer for their insightful suggestions and sincerely apology for our lack of representation of replicated experiments. We have amended this and as proposed by the reviewer have included the repeated experiments of all three methods in the Supplementary figures 1 and 2 and have changed the graphs to floating box plots that show the minimum, maximum and mean of the data.
We also thank the reviewer for their suggestion of increased exposure for the imaging for better detection of lower protein amounts. Although we had not represented it in the manuscript, we had used longer exposure times, especially with the iBind Flex blots (up to 62 sec) to better detect some of the proteins however this did not increase the sensitivity as such. We have included this information in the manuscript on page 4 line 179-180.
We also would like to thank the reviewer for bringing out attention to the missing information in the methods. We have added all the requested information as detailed below:
Line 75 - how many cells? Was this a dish or 6-well plate etc.
The following sentence has been added to page 2 line 79 “The cells plated in 6-well plates at 1.2 x 106 cells/well were transfected…”
Line 80 - was VEGF-B transfected using the same plasmid amount as the RBD constructs?
Yes the VEGF-B was transfected using the same plasmid amount of 2.5 ug, this has been added to the text on page 2 line 42
Line 88 - what is 1x sample buffer?
The sample buffer has been specified and a reference to it is given.
Line 93, and throughout - category numbers, and ideally RRID for antibodies should be provided. For example, Invitrogen sells many different goat anti-rabbit HRP antibodies, which one exactly did you use?
All the catalogue numbers and RRID information for the antibodies have been added on pages 2-3.
Line 110 - needs information about the secondary here too.
The catalogue numbers and RRID information for this antibody has been added.
Line 115 - needs information about the re-probing as currently sounds like the two primary antibodies were used together which is inappropriate.
The following sentence has been added (line 124-5) to rectify the ambiguity of the re-probing: ‘RBD and GAPDH were detected sequentially following stripping of the membranes after the first protein detection’
Line 234 - a comparison as to the mass of protein loaded in each technique would be useful to help compare between techniques.
In the JESS Simple Western™ the specific sample volume that is taken up into the columns is not known so precise protein amounts cannot be calculated. However, a range of the sample volume is known (30-50 nL) and so this has been added to the text and of the protein amount calculated to 0.375-0.625 ng (< 1ng). This information has been added to Lines 243-4 and 284.
Additional changes:
During the revision we found a mistake in the labeling of the protein amounts for the traditional Western blotting and iBind™ Flex immunobloting (10, 1 and 0.1 ug were indicated these should have been 10, 2.5 and 1 ug). We sincerely apology for this mistake which we now have corrected in the text and figure labels (figures 1 and 2)
We hope that we have addressed all the reviewers’ comments satisfactorily and the manuscript is now acceptable for publication in MDPI- Methods and Protocols.
Reviewer 3 Report
This manuscript compares the automated and traditional western blotting methods, revealing the advantages and disadvantages of conventional and automated WB. Automated WB reduces the handling time, improves sensitivity, and increases reproducibility compared to conventional WB. Importantly, it requires fewer samples for detection, suitable for limited or precious samples. The manuscript is well written. Here are some minor suggestions below:
1. The working principle of automatic WB equipment can be described in more general terms or illustrated visually with graphics to give the reader a clear picture.
2. For the example of the JESS Simple Western™ described in the fourth and fifth paragraphs of the results, the authors are recommended to set at the same concentration for conventional WB and iBind™ Flex as controls, in addition to stating that the method can detect lower concentrations of protein with high sensitivity.
3. The quality of the figures can be improved. For example, the error bars in the bar graph.
Author Response
First and foremost, we would like to sincerely thank the reviewers for their constructive comments which have improved this manuscript. We were encouraged by the reviewers’ positive assessment of our manuscript and have endeavored to address all their comments.
Specific responses to the points raised are listed below:
- The working principle of automatic WB equipment can be described in more general terms or illustrated visually with graphics to give the reader a clear picture.
We thank the reviewer for this comment. We have edited the text to describe the automated WB in more detail. The paragraph on Lines 63-73 now reads as follows:
‘In this instrument, capillaries replace the acrylamide gel and the transfer membrane, such that samples are loaded, size separated, and immunoblotted in the capillaries. The proteins in the samples separate in the capillaries by migrating through stacking and separation matrices and are immobilized to the capillary walls via photoactivated capture chemistry. This then allows for the immunoblotting to take place in the capillaries to detect specific immobilized proteins. The immunoblotting follows a standard regime of primary and (labelled) secondary Abs followed by detection system and imaging. Indeed, one of the advantages of JESS Simple Western™ is that the imaging of either a fluorescent tag or a chemiluminescence signal, as well as analysis of the images, are also completed within the run.’
- For the example of the JESS Simple Western™ described in the fourth and fifth paragraphs of the results, the authors are recommended to set at the same concentration for conventional WB and iBind™ Flex as controls, in addition to stating that the method can detect lower concentrations of protein with high sensitivity.
We thank the reviewer for this comment. We have equated the concentrations used in the JESS analysis to total protein amounts. This is a range since the absolute volume of sample taken up into the columns is not known but is normally 30-50nL. This information has been added to the results section, Lines 242-4, which now reads as the following:
‘The detection of protein at 12.5 ng/ µL equates to detection of total protein of 0.375-0.625 ng if taking into consideration the sample volume taken up into the columns is 30-50nL, this highlights the sensitivity of the automated WB over the other methods tested here.’
- The quality of the figures can be improved. For example, the error bars in the bar graph.
We sincerely apologize for the quality of the figures. We have changed most of the graphs to a floating box diagram to demonstrate the minimum, maximum and mean values of the data to better represent the variance. In addition, we have added the blot images from replicate experiments in supplementary figures 1 and 2 and have modified the text accordingly (pages 4-8).
Additional changes:
During the revision we found a mistake in the labeling of the protein amounts for the traditional WB and iBind™ Flex immunoblotting (10, 1 and 0.1 ug were indicated these should have been 10, 2.5 and 1 ug). We sincerely apology for this mistake which we now have corrected in the text and figure labels (figures 1 and 2)
We hope that we have addressed all the reviewers’ comments satisfactorily and the manuscript is now acceptable for publication in MDPI- Methods and Protocols.
Round 2
Reviewer 2 Report
I am happy with the changes made by the authors.